# The Relationship between Childhood Maltreatment and Risky Sexual Behaviors: A Meta-Analysis

**DOI:** 10.3390/ijerph16193666

**Published:** 2019-09-29

**Authors:** Zi-Yu Wang, Ming Hu, Tao-Lin Yu, Jun Yang

**Affiliations:** Department of Epidemiology and Health Statistics, XiangYa School of Public Health, Central South University, Changsha 410078, China; 176911011@csu.edu.cn (Z.-Y.W.); yutaolin1224@163.com (T.-L.Y.); 186911019@csu.edu.cn (J.Y.)

**Keywords:** childhood maltreatment, risky sexual behaviors, meta-analysis

## Abstract

Childhood maltreatment is associated with risky sexual behaviors (RSBs). Previous systematic reviews and meta-analysis focused only on the relationship between childhood sexual abuse and RSBs, thus the association between childhood maltreatment and RSBs has yet to be systematically and quantitatively reviewed. We aimed to provide a systematic meta-analysis exploring the effect of childhood maltreatment and its subtypes on subsequent RSBs in adolescence and adulthood. PubMed, Google Scholar, EMBASE, Medline were searched for qualified articles up to April 2019. We calculated the pooled risk estimates using either the random-effect model or fixed-effect model. The potential heterogeneity moderators were identified by subgroup and sensitivity analysis. Overall, childhood maltreatment was significantly associated with an early sexual debut (odds ratio (OR) = 2.22; 95% confidence interval (CI): 1.64–3.00), multiple sexual partners (OR = 2.22; 95% CI: 1.78–2.76), transactional sex (OR = 3.05; 95% CI: 1.92–4.86) and unprotected sex (OR = 1.59; 95% CI: 1.22-2.09). Additionally, different types of childhood maltreatment were also significantly associated with higher risk of RSBs. Relevant heterogeneity moderators have been identified by subgroup analysis. Sensitivity analysis yielded consistent results. Childhood maltreatment is significantly associated with risky sexual behaviors. The current meta-analysis indicates it is vital to protect children from any types of maltreatment and provide health education and support for maltreated individuals.

## 1. Introduction

Childhood maltreatment (CM) including neglect and emotional, sexual, and physical abuse has been recognized as a global public-health and social-welfare problem [1,2,3,4]. In high-income countries, about 4–16% of children suffered from physical abuse annually, and about 10% of children were neglected or emotionally abused [3]. It was reported that approximately 702,000 children were determined to be victims of childhood maltreatment nationally each year in the United States [5]. Among children aged 14–17 years in the USA, the estimated lifetime rate was 38.1% for childhood maltreatment, 18.1% for physical abuse, 23.9% for emotional abuse, 18.4% for neglect [2]. The estimated prevalence of childhood maltreatment among college students was 64.7% in China [6]. Research on the short- and long-term consequences of childhood maltreatment suggests that it substantially contributes to child mortality and morbidity, both internalizing and externalizing psychiatric disorders, criminal behaviors, substance use and risky sexual behavior in adolescence and adulthood [5,7,8,9,10].

Likewise, risky sexual behaviors (RSBs) in adolescence and adulthood represent a serious public health concern that may lead to poor reproductive health outcomes such as sexually transmitted diseases (STDs), human immunodeficiency virus (HIV), unintended pregnancy and pelvic inflammatory diseases [11,12]. Childhood maltreatment is associated with higher risk of RSBs including early sexual debut, higher number of partners, unprotected sexual intercourse, engagement in transactional sex and teen pregnancy [9,13,14,15]. Attachment theory, developmental traumatology and neurobiological dysregulation are common theoretical approaches to explain the mechanism linking CM and RSBs [16]. A robust body of literature connecting childhood maltreatment and risky sexual behavior has focused on outcomes for sexual abuse in particular [17,18,19,20]. The relationship between childhood sexual abuse (CSA) and subsequent RSBs in adolescence and adulthood has also been well established in both high-risk and general population [21,22,23,24,25]. However, there is equivocal evidence that other maltreatment forms predict risky sexual behavior as strongly as CSA [26,27,28]. A prospective study with a 30-year follow-up documented that physical abuse, sexual abuse, and neglect increased risk of early sexual contact and prostitution [29]. Furthermore, according to another prospective birth cohort study, the association between sexual abuse, physical abuse, emotional abuse, and neglect and early age at sexual intercourse was independent after adjusting for confounding factors [9]. Additionally, in that study, neglect was found to be also associated with multiple sexual partners and emotional abuse with higher rates of miscarriage.

Following these results, when exploring the effects of childhood maltreatment on risky sexual behaviors, different subtypes need to be considered. Previous systematic reviews and meta-analysis [30,31,32,33] focused only on the relationship between sexual abuse and risky sexual behaviors, thus the association between childhood maltreatment and RSBs has yet to be systematically and quantitatively reviewed. A meta-analysis of nine articles indicated that sexually abused boys compared with non-abused boys were at a higher risk of having unprotected intercourse (OR = 1.91), having multiple sexual partners (OR = 2.91) and causing a pregnancy (OR = 4.81) [31]. Another meta-analysis demonstrated that childhood sexual abuse was associated with a 1.59 times greater likelihood of RSBs, and the significant association did not vary by gender. Based on previous research, we hypothesized that childhood maltreatment, as well as specific forms of childhood maltreatment, all increased the risk of having RSBs in adolescence and adulthood [30]. The aims of the current meta-analysis were twofold. First, we aimed to provide a systematic meta-analysis exploring the effect of childhood maltreatment and its subtypes on subsequent RSBs in adolescence and adulthood. Second, we identified potential heterogeneity moderators by subgroup and sensitivity analysis.

## 2. Materials and Methods

### 2.1. Literature Search 

We conducted the present meta-analysis strictly following the proposed PRISMA (Preferred Reporting Items for Systematic Reviews and Meta-Analyses Protocols) statement. A systematic search of relevant articles in PubMed, Google Scholar, EMBASE, Medline was undertaken by two researchers. The following search terms were applied: “adolescent risky sexual behaviors”, “adult risky sexual behaviors”, “adolescent reproductive health outcomes”, “adult reproductive health outcomes”, “childhood maltreatment”, “child abuse”, “child adversity”, “child trauma”, “child neglect”, “child physical abuse”, “child emotional abuse”. We limited our searches to studies that were published between 1990 and 2018 and written in English, this time frame was chosen because the awareness of childhood maltreatment and its potential consequences was heightened globally during the time. We also manually searched review articles and reference lists of relevant studies. 

### 2.2. Exposure and Outcome Indicators of Interest 

The exposure of interest was childhood maltreatment, which was further categorized into childhood sexual abuse (CSA), childhood physical abuse (CPA), childhood emotional abuse (CEA) and childhood neglect (CN). The outcomes of interest were risky sexual behaviors, including at least one of the following outcomes: early sexual debut, multiple sexual partners, transactional sex or unprotected sex. In our study, we focused not only on the association between total childhood maltreatment and any form of RSBs, but the association between subtypes of childhood maltreatment and any form of RSBs.

### 2.3. Inclusion Criteria

We identified studies satisfying the following criteria: (1) population-based observational quantitative studies; (2) studies reported subtype(s) of childhood maltreatment such as childhood sexual, physical or emotional abuse, and/or neglect; (3) studies reported the associations between a history of childhood maltreatment and any form of risky sexual behavior; (4) studies included participants of both genders and reported the outcomes of the total sample; and (5) studies reported the relative risks (RRs) and odd ratios (ORs), with corresponding 95% confidence intervals (CIs) (or data to calculate them).

### 2.4. Data Extraction and Quality Assessment

Two independent researchers (WZY, HM) extracted data and assessed study quality. The two researchers agreed upon any discrepancies through discussion and if necessary referred the issue to a third researcher. Using a study-designed standardized form, we extracted the available information as follows: first author, year of publication, study design, study country, total sample, subtype of childhood maltreatment and quality score. We used a ten-point Newcastle–Ottawa Scale (NOS) to assess the quality of included studies in the present review. Each study was judged on three broad perspectives: the selection of the study groups; the comparability of the groups; and the ascertainment of either the exposure or outcome of interest. When the study wins seven or more stars, it is considered of higher methodologic quality.

### 2.5. Statistical Analysis

We abstracted one effect size per outcome from each primary study. Odds ratios (ORs) with corresponding 95% confidence intervals (CIs) were used given the majority of the studies were cross-sectional. When evidence of heterogeneity existed across studies, we used random-effects models to calculate the pooled OR and the corresponding 95% CI, whereas fixed-effects models were used. When multiple types of childhood maltreatment were reported in the same study, the effect values of the relationship between different types of childhood maltreatment and the risk of having any form of risky sexual behaviors were combined with statistical software. Homogeneity of effect size across studies was tested by using the *Q* statistics at the *P* < 0.10 level of significance. The *I*^2^ statistic, which is a quantitative measure of inconsistency across studies, was also calculated (significance level at *I*^2^ > 50%).

A series of sub-group analyses according to geographic region, assessment method of CM, study design, study sample, study quality and year of publication were conducted to assess the potential effect modification of these variables on outcomes. We also conducted a sensitivity analysis by removing individual studies one at a time and recalculated a pooled OR to examine whether any one study overtly influenced the pooled effect size. We examined the possibility of publication bias, using Egger’s test and Begg’s rank correlation test (significance level at *P* < 0.10). All statistical analyses were performed using Stata version 13.0 and Review Manager version 5.3.

## 3. Results

### 3.1. Search Results and Study Characteristics

We initially identified 4100 potentially eligible records from the electronic databases. Two additional articles were found from reference lists. Finally, 19 eligible studies were identified (Figure 1). The characteristics of included literatures, which involved 74,557 participants and were published between 1990 and 2018, were summarized in Table 1. Among the 19 studies included, 11 (57.9%) studies were conducted in America, 3 (15.8%) in Asia, 1 (5.3%) in Oceania, 1 (5.3%) in Europe and 3 (15.8%) in Africa. The majority, 73.7% (n = 14), of included studies were cross-sectional whereas 26.3% (n = 5) were cohort studies. 8 (42.1%) studies were considered of higher methodological quality, achieving a quality score ≥ 7 out of 9. The outcomes of early sexual debut in 6 studies were reported, multiple sexual partners in 15 studies, transactional sex in 8 studies, and unprotected sex in 9 studies. The number of studies reporting the exposure of subtypes of CM were as follows: 5 CSA, 4 CPA, 2 CEA, and 3 CN in early sexual debut; 12 CSA, 8 CPA, 4 CEA, and 5 CN in multiple sexual partners; 8 CSA, 2 CPA, and 1 CN in transactional sex; 9 CSA and 2 CPA in unprotected sex. 

### 3.2. Childhood Maltreatment and Risk of Having an Early Sexual Debut

Overall, childhood maltreatment was associated with a 2.22 times greater likelihood of having an early sexual debut (OR = 2.22; 95% CI: 1.64–3.00). However, significant heterogeneity was found (*I*^2^ = 81%, *P* < 0.01) (Figure 2). With regard to specific CM subtypes, the present meta-analysis showed the risk of having an early sexual debut was significantly increased in adolescents and adults who experienced CSA (OR = 3.59; 95%CI: 1.95–6.62), CPA (OR = 1.54; 95%CI: 0.94–2.51), CEA (OR = 1.72; 95%CI: 1.24–2.39), and CN (OR = 1.58 95%CI: 1.03–2.43). Nevertheless, there was substantial heterogeneity except for CEA (*I*^2^ = 9%; *P* = 0.29) (Figure 3).

### 3.3. Childhood Maltreatment and Risk of Having Multiple Sexual Partners 

The results of pooled estimates for the association between childhood maltreatment and having multiple sexual partners was 2.22 (95% CI: 1.78–2.76). Yet, substantial heterogeneity was found (*I*^2^ = 91%, *P* < 0.01) (Figure 2). With regard to specific CM subtypes, the present meta-analysis showed those who experienced CSA (OR = 2.63; 95% CI: 1.95–3.54), CPA (OR = 1.54; 95% CI: 1.28–1.86), CEA (OR = 1.74; 95% CI: 1.22–2.49) or CN (OR = 1.44; 95% CI: 1.08–1.92) had a significant higher risk of having multiple sexual partners. However, substantial heterogeneity was found in all groups, with *I*^2^ as 84%, 58%, 73%, 79%, and the *P* value was <0.01, 0.02, 0.01 and <0.01, respectively (Figure 4).

### 3.4. Childhood Maltreatment and Risk of Having Transactional Sex 

The overall pooled OR for the association between childhood maltreatment and having transactional sex was 3.05 (95% CI: 1.92–4.86). However, substantial heterogeneity was found (*I*^2^ = 84%; *P* < 0.01) (Figure 2). With regard to specific CM subtypes, the present meta-analysis showed CSA (OR = 3.24; 95% CI: 2.10–4.99) or CPA (OR = 1.86; 95% CI: 1.20–2.89) was associated with higher risk of having transactional sex. Evidence of heterogeneity was observed for CSA (*I*^2^ = 74%; *P* < 0.01), but not for CPA (*I*^2^ = 20%; *P* = 0.26) (Figure 5).

### 3.5. Childhood Maltreatment and Risk of Having Unprotected Sex 

The pooled OR for the association between childhood maltreatment and having unprotected sex was 1.59 (95% CI: 1.22–2.09). Nevertheless, substantial heterogeneity was found (*I*^2^ = 88%; *P* < 0.01) (Figure 2). With regard to specific CM subtypes, the present meta-analysis showed individuals experiencing CSA (OR = 1.57; 95% CI: 1.16–2.13) or CPA (OR = 1.25; 95% CI: 1.08–1.44) had a higher risk of having unprotected sex. Significant heterogeneity was found in CSA (*I*^2^ = 88%; *P* < 0.01), but not in CPA (*I*^2^ = 38%; *P* = 0.20) (Figure 6).

### 3.6. Subgroup Analysis 

Subgroup analyses for risk estimates between childhood maltreatment and risky sexual behaviors are summarized in Table 2. Overall, a significantly increased risk of having RSBs was found in most of subgroups. For risk estimates between childhood maltreatment and having an early sexual debut, after subgroup analysis, study design (test for subgroup difference (TSD): *I*^2^ = 54.4%) was identified as the relevant heterogeneity moderator. However, the risk of having an early sexual debut was not statistically different for study design (*P*= 0.14). For risk estimates between childhood maltreatment and having multiple sexual partners, after subgroup analysis, study design (TSD: *I*^2^ = 90.6%), assessment method of CM (TSD: *I*^2^ = 67.1%) and geographic region (TSD: *I*^2^ = 2.8%) were identified as the first three most relevant heterogeneity moderators. The risk of having multiple sexual partners was statistically different for study design (*P* < 0.01). For risk estimates between childhood maltreatment and having transactional sex, after subgroup analysis, geographic region (TSD: *I*^2^ = 85.5%), study design (TSD: *I*^2^ = 1.1%) and quality score (TSD: *I*^2^ = 1.1%) were identified as the first three most relevant heterogeneity moderators. The risk of having transactional sex was statistically different for geographic region (*P* < 0.01). For risk estimates between childhood maltreatment and having unprotected sex, after subgroup analysis, study design (TSD: *I*^2^ = 87.3%), study sample (TSD: *I*^2^ = 71.7%) and geographic region (TSD: *I*^2^ = 59.2%) were identified as the first three most relevant heterogeneity moderators. The risk of having unprotected sex was statistically different for study design (*P* < 0.01).

### 3.7. Sensitivity Analysis 

Sensitivity analyses were conducted to assess the stability of the results. We removed individual studies one at a time and recalculated a pooled OR to examine whether any one study overtly influenced the pooled effect size. Sensitivity analyses showed exclusion of any single study at a time did not materially alter the overall pooled estimates. For risk estimates between childhood maltreatment and having an early sexual debut, the pooled OR ranged from 1.91 (1.54–2.37) to 2.28 (1.55–3.34); for risk estimates between childhood maltreatment and having multiple sexual partners, the pooled OR ranged from 2.02 (1.66–2.46) to 2.33 (1.89–2.88); for risk estimates between childhood maltreatment and having transactional sex, the pooled OR ranged from 2.46 (1.80–3.34) to 3.46 (2.21–5.43); for risk estimates between childhood maltreatment and having unprotected sex, the pooled OR ranged from 1.44 (95% CI: 1.13–1.85) to 1.70 (95% CI: 1.27–2.29).

### 3.8. Publication Bias 

Both the Begg’s rank correlation test (*P* = 0.024) and Egger’s test (*P* = 0.020) indicated evidence of publication bias for risk estimates between CM and having an early sexual debut. Evidence of publication bias was observed by Egger’s test (*P* = 0.008), but not observed by Begg’s rank correlation test (*P* = 0.075) for risk estimates between CM and having multiple sexual partners. No indication of publication bias was observed by Begg’s rank correlation test (*P* = 0.174 for transactional sex; *P* = 0.522 for unprotected sex) or Egger’s test (*P* = 0.741 for transactional sex; *P* = 0.329 for unprotected sex) for risk estimates between CM and having transactional sex and having unprotected sex.

## 4. Discussion

Our meta-analysis yielded the following main findings. First, overall, childhood maltreatment was significantly associated with an increased risk of having an early sexual debut (OR = 2.22), multiple sexual partners (OR = 2.22), transactional sex (OR = 3.05) and unprotected sex (OR = 1.59). Second, different types of childhood maltreatment were also significantly associated with an increased risk of RSBs. For example, CSA (OR = 3.59), CEA (OR = 1.72) or CN(OR = 1.58) was significantly associated with risk of having an early sexual debut; CSA (OR = 2.63), CPA (OR = 1.54), CEA (OR = 1.74) and CN (OR = 1.44) also increased the risk of having multiple sexual partners; individuals experiencing CSA and CPA had a significantly higher risk of having transactional sex (OR_CSA_ = 3.24, OR_CPA_ = 1.86) and having unprotected sex (OR_CSA_ = 1.57, OR_CPA_ = 1.25). Third, the association between childhood maltreatment and higher risk of RSBs still existed after subgroup and sensitivity analyses, indicating our results were stable and credible.

So far, there are four existing reviews [30,31,32,33] that evaluated the association of childhood maltreatment with RSBs. However, these reviews exclusively focused on CSA and little is known about the effect of other types of childhood maltreatment on RSBs. Our study has important strengths compared with previous meta-analyses. To our knowledge, our study was the first meta-analysis to examine the relationship between childhood maltreatment (which was further categorized into CSA, CPA, CEA and CN) and RSBs. Additionally, sufficient numbers of studies reported the same individual outcome to be able to meta-analyze RSB outcomes separately. In our study, more than half of studies had a large sample size (>1000); 45.0% studies were considered of higher methodological quality and these high-quality studies contributed most study participants. With the accumulating evidence and enlarged sample size, we have enhanced statistical power to provide more precise and reliable risk estimates. To handle the heterogeneity, sub-group analyses according to geographic region, assessment method of CM, study design, study sample, study quality and year of publication were conducted in our study. The most relevant heterogeneity moderators were identified by subgroup analysis. We found that the association between childhood maltreatment and risk of RSBs remains statistically significant after subgroup and sensitivity analyses.

The exact mechanisms involved in the association between childhood maltreatment and RSBs are still unclear, but attachment theory could provide further insight into them [16]. Evidences show childhood maltreatment may lead to an insecure attachment that then plays a role in risky sexual behaviors [16,45,46]. From a theoretical perspective, maltreated victims are more likely to engage unwanted SRBs to avoid disapproval and rejection or have casual sexual behaviors to inhibit the development of deep emotional attachment. Consistent with this theory, individuals with adverse childhood neglect may increase the likelihood of having multiple sexual partners, to evade contact and closeness with others [47]. However, maltreated individuals may also use sex as means for securing affection and intimacy, potentially leading to earlier sexual debut [29,30].

In addition, trauma symptoms appear to be an important explanation for the link [48]. Childhood maltreatment may cause a range of internalizing problems in childhood and cause trauma symptoms by distorting the child’s concept of self and others, world view, or affective capacities, making victims vulnerable to RSBs during adolescence or adulthood [9,49,50]. Evidence shows shame, feelings of betrayal or powerlessness components of trauma symptoms in individuals experiencing CEA and CSA may reduce awareness of protective behaviors, in particular contraceptive use [51,52,53]. Furthermore, individuals with maltreated histories may use alcohol to cope with these trauma symptoms, and previous studies have proposed that alcohol use may lead to high risk of RSBs [47,54]. Alexithymia was also confirmed as a mediator of the relationship between CM and subsequent risky sexual behaviors. Individuals who have experienced childhood maltreatment have emotional deficiencies in understanding and expressing emotions and may suffer from social isolation and poor interpersonal relationships [55]. Thus, they may be likely to seek immediate interpersonal interactions through risky sexual behavior. 

Of course, there are other potential explanations for the association between CM and risk of RSBs. Recent studies on the neurobiological perspective show that CM is associated with reduced medial prefrontal cortex (mPFC) gray matter and the alterations may possibly enhance sexual decision making and risky sexual behaviors [56,57]. Moreover, maltreated children may have higher rates of sexual preoccupation, that may result in more sexual stimulation, particularly an early sexual debut [58,59]. Finally, ADHD symptoms in maltreated individuals were reported to be associated with early sexual debut and multiple partners [5,60].

The present meta-analysis has some limitations. First, the substantial heterogeneity among studies for the association between childhood maltreatment and RSBs were observed. However, this heterogeneity is not surprising given the differences in sociodemographic status and methodology. In our study, we conducted subgroup analyses to explain some extent of this heterogeneity and have identified the main heterogeneity moderators, including study design, assessment method of CM, geographic region, study sample and quality score. Although heterogeneity was still observed after subgroup or sensitivity analyses, the association between childhood maltreatment and RSBs still existed and the results were stable. However, many primary studies in our meta-analysis did not provide information on other variables which could explain some of the variance, such as duration of maltreatment, severity of maltreatment, the age of onset, perpetrator and comorbid disorders. 

Second, the majority of studies enrolled in the analyses were cross-sectional studies and retrospective cohort studies with the possibility of recall bias in self-reported childhood maltreatment and RSBs, thereby under- or over-estimating the strength of the association. However, evidence indicated the recall of childhood maltreatment remained fairly accurate over a long time. Third, discrepancies also existed in the definition of both CM (subtypes including CSA, CPA, CEA and CN) and RSBs, and the measurement of the constructs, frequency, and time frames of the outcomes, therefore our findings should be interpreted with caution. Fourth, many primary studies focused on school and community samples and few studies included street-involved youth and homeless individuals, thus our study may not be generalizable to other adolescent and adult populations. Finally, our study only included studies published in English, additional research in other populations is warranted to generalize the findings.

Our study has important implications both for future research and prevention. To reduce the consequences of CM, we should be aware that not only child sexual abuse leads to RSBs, all forms of CM and other specific forms of CM (CPA, CEA and CN) also contribute to RSBs. It is crucial to carry out research to clarify the unique effects of different forms of CM on RSBs and provide the evidence to take intervention measures in the future. However, out of 19 studies, only one of the included studies is from Europe, which indicates more sound research is supposed to be conducted in this valuable field. With regard to prevention, our findings highlight the need for effective programs and policies to screen for childhood maltreatment and provide more medical and reproductive health services for people who have experienced child maltreatment.

## 5. Conclusions

In conclusion, the findings of our meta-analysis indicate that childhood maltreatment and its subtypes increase the risk of RSBs, although the potential bias and evidence of heterogeneity should be carefully considered. The current meta-analysis indicates that it is necessary to screen for maltreated history in childhood and provide health education and support for maltreated individuals, which have crucial implications for clinical practice and interventions.

## Figures and Tables

**Figure 1 ijerph-16-03666-f001:**
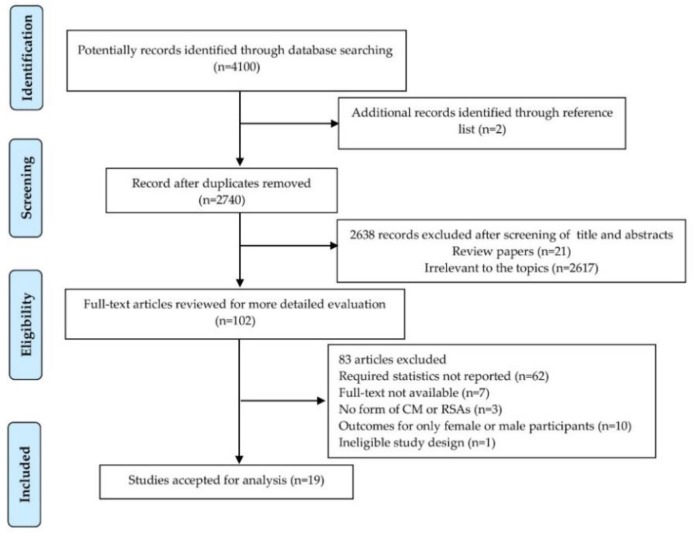
PRISMA (Preferred Reporting Items for Systematic Reviews and Meta-Analyses Protocols) statement flow diagram.

**Figure 2 ijerph-16-03666-f002:**
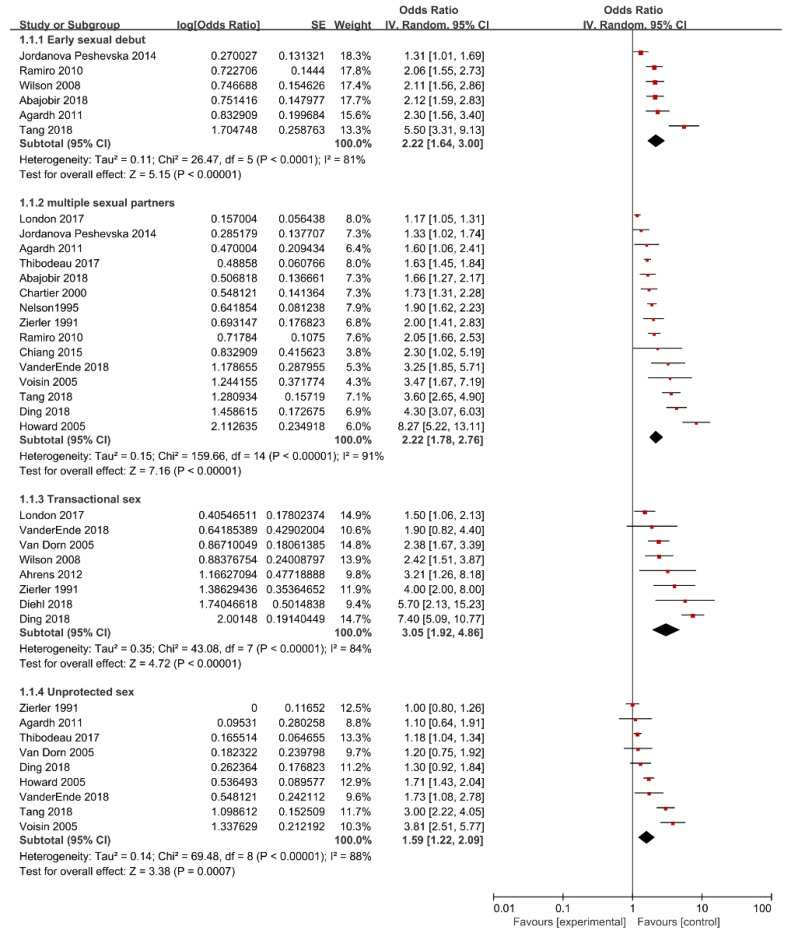
Forest plot for childhood maltreatment and risk of risky sexual behaviors (RSBs).

**Figure 3 ijerph-16-03666-f003:**
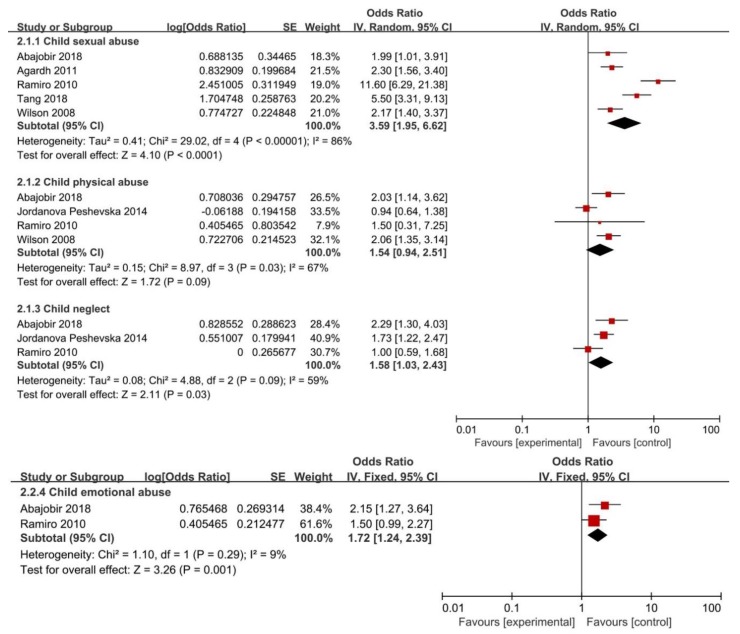
Forest plot for subtypes of childhood maltreatment and risk of having an early sexual debut.

**Figure 4 ijerph-16-03666-f004:**
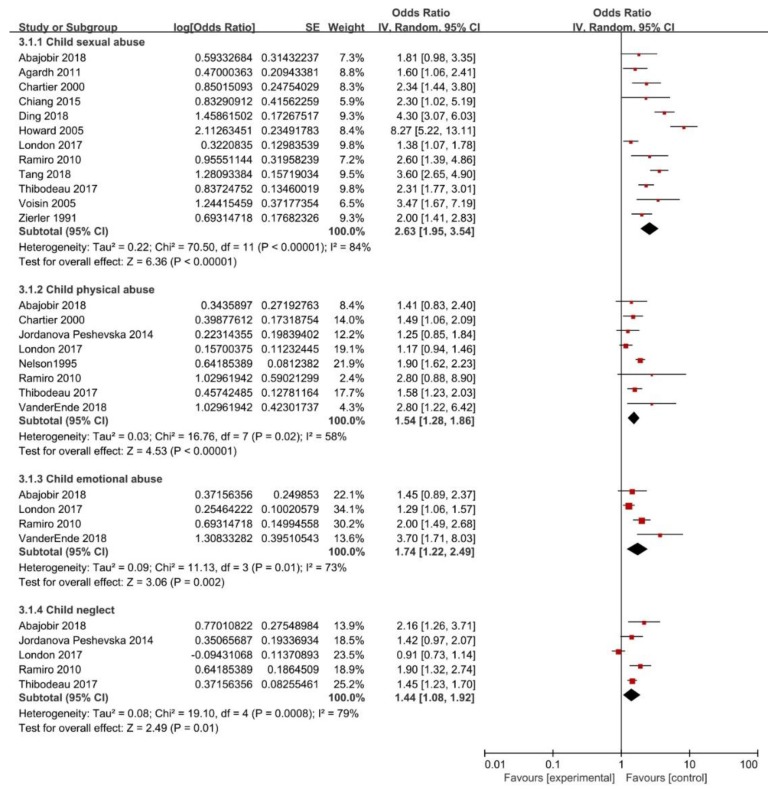
Forest plot for subtypes of childhood maltreatment and risk of having multiple sexual partners.

**Figure 5 ijerph-16-03666-f005:**
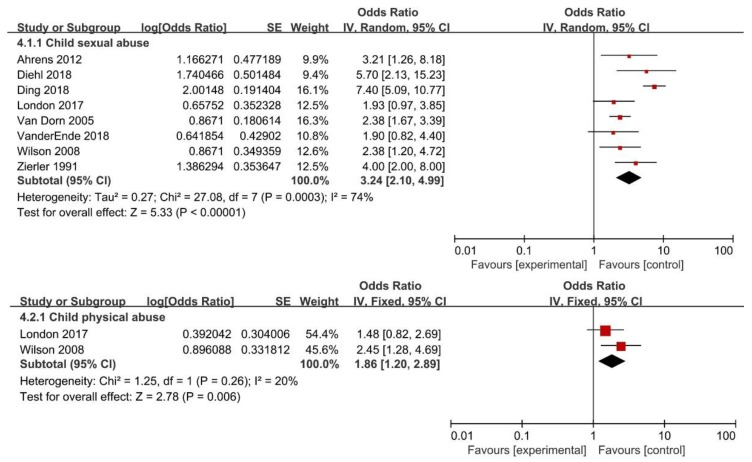
Forest plot for subtypes of childhood maltreatment and risk of having transactional sex.

**Figure 6 ijerph-16-03666-f006:**
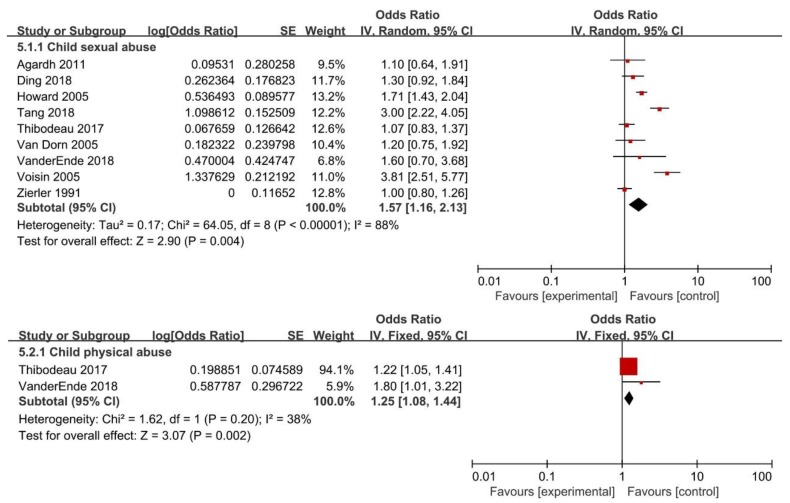
Forest plot for subtypes of childhood maltreatment and risk of having unprotected sex.

**Table 1 ijerph-16-03666-t001:** Summary information of studies included in the meta-analysis.

Study Author(s) (Year)	Geographic Region	Sample Size	Assessment of CM	Study Design	Study Sample	Quality Sore	Form(s) of CM
Abajobir (2018) [9]	Oceania	3081	Substantiated record	Cohort study	Other	8	CSA, CPA, CEA, CN
Ahrens (2012) [34]	America	574	SR	Cohort study	Other	6	CSA
Chartier (2009) [20]	America	8116	SR	Cross-sectional study	Other	6	CSA, CPA
Chiang (2015) [35]	Africa	3739	SR	Cross-sectional study	Other	6	CSA
Ding (2018) [36]	Asia	4974	SR	Cross-sectional study	Other	6	CSA
Jordanova Peshevska (2014) [37]	America	1277	SR	Cross-sectional study	School sample	7	CPA, CN
Thibodeau (2017) [38]	America	1940	SR	Cross-sectional study	School sample	7	CSA, CPA CN
Ramiro (2010) [13]	Asia	1068	SR	Cross-sectional study	Other	6	CSA, CPA CEA, CN
Nelson (1995) [27]	America	1957	SR	Cross-sectional study	School sample	5	CPA
VanderEnde (2018) [15]	Africa	610	SR	Cross-sectional study	Other	6	CSA, CPA CEA
London (2017) [39]	America	12288	SR	Cohort study	Other	7	CSA, CPA CEA, CN
Diehl (2018) [23]	America	134	SR	Cross-sectional study	Other	5	CSA
Agardh (2011) [40]	Africa	980	SR	Cross-sectional study	School sample	5	CSA
Tang (2018) [24]	Asia	17966	SR	Cross-sectional study	School sample	7	CSA
Van Dorn (2005) [41]	America	609	SR	Cross-sectional study	Other	5	CSA
Voisin (2005) [42]	America	409	SR	Cross-sectional study	School sample	6	CSA
Howard (2005) [43]	America	13601	SR	Cross-sectional study	School sample	7	CSA
Wilson (2008) [29]	America	1070	Substantiated record	Cohort study	Other	9	CSA, CPA
Zierler (1991) [44]	Europe	164	SR	Cohort study	Other	7	CSA

SR: self-reported; CM: childhood maltreatment; CSA: childhood sexual abuse; CPA: childhood physical abuse; CEA: childhood emotional abuse; CN: childhood neglect.

**Table 2 ijerph-16-03666-t002:** Subgroup analysis of association between childhood maltreatment and risky sexual behaviors.

Subgroup	Early Sexual Debut	Multiple Sexual Partners	Transactional Sex	Unprotected Sex
**Geographic Region**	TSD: *I*^2^ = 0% *P* = 0.38	TSD: *I*^2^ = 2.8% *P* = 0.38	TSD: *I*^2^ = 85.5% *P* < 0.01	TSD: *I*^2^ = 59.2% *P* = 0.06
America	1.65 (1.04, 2.63) n = 2	2.00 (1.49, 2.69) n = 7	2.29 (1.55, 3.38) n = 4	1.71 (1.14, 2.55) n = 4
	*I*^2^ = 82% *P* = 0.02	*I*^2^ = 93% *P* < 0.01	*I*^2^ = 64% *P* = 0.04	*I*^2^ = 91% *P* < 0.01
Asia	3.27 (1.26, 8.60) n = 2	3.12 (1.94, 5.03) n = 3	7.40 (5.09, 10.77) n = 1	1.98 (0.87, 4.50) n = 2
	*I*^2^ = 91% *P* < 0.01	*I*^2^ = 88% *P* < 0.01	Not applicable	*I*^2^ = 92% *P* < 0.01
Europe	Not applicable	2.00 (1.41, 2.83) n = 1	4.00 (2.00, 8.00) n = 1	1.00 (0.80, 1.26) n = 1
		Not applicable	Not applicable	Not applicable
Other	2.18 (1.73, 2.75) n = 2	1.83 (1.49, 2.24) n = 4	2.40 (1.29, 4.49) n = 2	1.43 (1.00, 2.04) n = 2
	*I*^2^ = 0% *P* = 0.74	*I*^2^ = 42% *P* = 0.16	*I*^2^ = 0% *P* = 0.41	*I*^2^ = 33% *P* = 0.22
**Assessment of CM**	TSD: *I*^2^ = 0% *P* = 0.72	TSD: *I*^2^ = 67.1% *P* = 0.08	TSD: *I*^2^ = 0% *P* = 0.56	Not applicable
Substantiated records	2.12 (1.72, 2.61) n = 2	1.66 (1.27, 2.17) n = 1	2.56 (1.68, 3.90) n = 2	
	*I*^2^ = 0% *P* = 0.98	Not applicable	*I*^2^ = 0% *P* = 0.60	Not applicable
Self-report	2.33 (1.41, 3.85) n = 4	2.28 (1.80, 2.88) n = 14	3.18 (1.75, 5.79) n = 6	1.59 (1.22, 2.09) n = 9
	*I*^2^ = 88% *P* < 0.01	*I*^2^ = 92% *P* < 0.01	*I*^2^ = 88% *P* < 0.01	*I*^2^ = 88% *P* < 0.01
**Study design**	TSD: *I*^2^ = 54.4% *P* = 0.14	TSD: *I*^2^ = 90.6% *P* < 0.01	TSD: *I*^2^ = 1.1% *P* = 0.31	TSD: *I*^2^ = 87.3% *P* < 0.01
Cross-sectional study	2.86 (1.69, 4.86) n = 3	2.62 (2.03, 3.38) n = 11	3.74 (1.79, 7.83) n = 4	1.70 (1.27, 2.29) n = 8
	*I*^2^ = 82% *P* < 0.01	*I*^2^ = 89% *P* < 0.01	*I*^2^ = 86% *P* < 0.01	*I*^2^ = 88% *P* < 0.01
Cohort study	1.79 (1.29, 2.48) n = 3	1.46 (1.15, 1.86) n = 4	2.39 (1.51, 3.78) n = 4	1.00 (0.80, 1.26) n = 1
	*I*^2^ = 75% *P* = 0.02	*I*^2^ = 76% *P* < 0.01	*I*^2^ = 63% *P* = 0.05	Not applicable
**Study sample**	TSD: *I*^2^ = 0% *P* = 0.67	TSD: *I*^2^ = 0% *P* = 0.53	Not applicable	TSD: *I*^2^ = 71.7% *P* = 0.06
School sample	2.49 (1.14, 5.42) n = 3	2.42 (1.72, 3.38) n = 7		1.91 (1.26, 2.89) n = 5
	*I*^2^ = 92% *P* < 0.01	*I*^2^ = 92% *P* < 0.01	Not applicable	*I*^2^ = 93% *P* < 0.01
Other	2.10 (1.77, 2.48) n = 3	2.08 (1.51, 2.87) n = 8	3.05 (1.92, 4.86) n = 8	1.16 (0.98, 1.37) n = 4
	*I*^2^ = 0% *P* = 0.99	*I*^2^ = 91% *P* < 0.01	*I*^2^ = 84% *P* < 0.01	*I*^2^ = 37% *P* = 0.19
**Quality score**	TSD: *I*^2^ = 0% *P* = 0.80	TSD: *I*^2^ = 0% *P* = 0.67	TSD: *I*^2^ = 1.1% *P* = 0.31	TSD: *I*^2^ = 0% *P* = 0.85
≥7	2.02 (1.43, 2.87) n = 4	2.11 (1.51, 2.96) n = 7	2.39 (1.51, 3.78) n = 4	1.55 (1.06, 2.25) n = 4
	*I*^2^ = 88% *P* < 0.01	*I*^2^ = 94% *P* < 0.01	*I*^2^ = 66% *P* = 0.02	*I*^2^ = 93% *P* < 0.01
<7	2.14 (1.70, 2.69) n = 2	2.31 (1.83, 2.92) n = 8	3.74 (1.79, 7.83) n = 4	1.64 (1.04, 2.59) n = 5
	*I*^2^ = 0% *P* = 0.65	*I*^2^ = 74% *P* < 0.01	*I*^2^ = 86% *P* < 0.01	*I*^2^ = 81% *P* < 0.01
**Year of publication**	TSD: *I*^2^ = 0% *P* = 0.74	TSD: *I*^2^ = 0% *P* = 0.56	TSD: *I*^2^ = 0% *P* = 0.59	TSD: *I*^2^ = 0% *P* = 0.80
<2013	2.13 (1.77, 2.56) n = 3	2.38(1.76, 3.21) n = 7	2.58 (1.98, 3.35) n = 3	1.54 (1.01, 2.35) n = 5
	*I*^2^ = 0% *P* = 0.90	*I*^2^ = 85% *P* < 0.01	*I*^2^ = 0% *P* = 0.40	*I*^2^ = 89% *P* < 0.01
≥2013	2.40 (1.21, 4.75) n = 3	2.09 (1.53, 2.85) n = 8	3.27 (1.43, 7.47) n = 5	1.67 (1.05, 2.66) n = 4
	*I*^2^ = 92% *P* < 0.01	*I*^2^ = 93% *P* < 0.01	*I*^2^ = 90% *P* < 0.01	*I*^2^ = 91% *P* < 0.01

CM: childhood maltreatment; TSD: test for subgroup differences.

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
