# Peer review of "The Relationship between Childhood Maltreatment and Risky Sexual Behaviors: A Meta-Analysis"

_ijerph, 2019, doi:10.3390/ijerph16193666_

Round 1
Reviewer 1 Report
I think the authors have made a great effort in explaining the background, and their study implies a new step towards the understanding of the relationship between child maltreatment and risky sexual behaviors. In my opinion, the manuscript is well written, properly documented, justified and organized, results are well presented and conclusions are directly derived from the results.
Before accepting it for publication I would like the authors to address some minor concerns. I am making mostly minor remarks and it is my hope that they will help to improve the manuscript.
1) In the literature search subsection authors stated that the studies searched were published between 1990 and 2018. Authors should include a rationale to explain the relevance of that time frame.
2) In the limitations section authors explain that all the studies analyzed were in English. It would be good to include this information in the Inclusion criteria subsection.
3) Authors used the Newcastle-Ottawa Scale to assess the quality of included studies. It would be good in they briefly explain what types of indicators measure the scale to assess the quality.
4) In page 3, line 125 there is a typo when authors indicate that 3 studies were conducted in Africa.
5) I would like to see more elaboration regarding the implication of the meta-analyses for future research.
6) Regarding implications for practice. Is there any program actually working with maltreated children in order to prevent risky sexual behaviors?
Author Response
Thank you for your serious and constructive comments on our manuscript. According to your suggestion, the manuscript has been revised as a letter to editor. The revisions we have made are as follows:
Point 1: In the literature search subsection authors stated that the studies searched were published between 1990 and 2018. Authors should include a rationale to explain the relevance of that time frame.
Response 1: Thank you for your constructive and helpful suggestion. In the revised paper, the relevant explanation has been mentioned in the Literature search part, in page 2, line 84 to 86.
Point 2: In the limitations section authors explain that all the studies analyzed were in English. It would be good to include this information in the Inclusion criteria subsection.
Response 2:Thank you for your constructive and helpful suggestion. We have included this information in the Literature search part, in page 2, line 84.
Point 3: Authors used the Newcastle-Ottawa Scale to assess the quality of included studies. It would be good in they briefly explain what types of indicators measure the scale to assess the quality.
Response 3: Thank you for your constructive and helpful suggestion. In the revised paper, the relevant explanation has been mentioned in the Data extraction and quality assessment part, in page 3, line 111 to 114.
Point 4: In page 3, line 125 there is a typo when authors indicate that 3 studies were conducted in Africa.
Response 4: Thank you for your constructive and helpful suggestion. The relevant typo have been revised in the paper.
Point 5: I would like to see more elaboration regarding the implication of the meta-analyses for future research.
Response 5: Thank you for your constructive and helpful suggestion. In the revised paper, the implication of the meta-analyses for future research has been mentioned in the Discussion part, in page 14, line 319 to 325.
Point 6: Regarding implications for practice. Is there any program actually working with maltreated children in order to prevent risky sexual behaviors?
Response 6: Thank you for your constructive and helpful suggestion. In the revised paper, we mentioned the information in Discussion part, in page 14, line 325 to 327.

Reviewer 2 Report
The introduction provides significant statistics that provide sufficient information about the problem of childhood maltreatment. The negative outcomes are also clearly expressed in the introduction.
However, the literature review is a bit too brief, and needs further development. I would like to see more development in the discussion of previous lit reviews/meta-analysis and a discussion of their effect sizes, or more detail about what we already know.
The selection criteria was well described for the meta-analysis. The authors provided sufficient information about each of the articles included, and the tables illustrated the information from the text.
The analyses was conducted professionally and with accuracy.
The discussion section also illustrates the meaning of the main findings in the context of previous research. There is also a discussion of the theoretical value of the findings, which can enhance future scholarship relevant to childhood maltreatment.
As a whole, the article is well-written, organized, and well-supported throughout.
Thus, I only had very minor concerns about the introduction.
Author Response
Thank you for your serious and constructive comments on our manuscript. According to your suggestion, the manuscript has been revised as a letter to editor. The revisions we have made are as follows:
Point 1: The literature review is a bit too brief, and needs further development. I would like to see more development in the discussion of previous lit reviews/meta-analysis and a discussion of their effect sizes, or more detail about what we already know.
Response 1: Thank you for your constructive and helpful suggestion. In the revised paper, the relevant information has been mentioned in page 2, line 64 to 70.

Reviewer 3 Report
The work entitled “The relationship between childhood maltreatment and risky sexual behaviors: A Meta-Analysis” is of great interest. The research contains new scientific knowledge. The data is presented in a clear and easy-to-understand fashion. I have some comments to make that should be addressed before I recommend this manuscript for publication.
The introduction section may benefit from more information about risky sexual behaviors in both adolescence and adult life. For instance, information about possible causes and theories about may help readers to understand the relevance of the study and could help authors to elaborate the discussion section.
With this regard, the objectives subsection in the introduction should include hypothesis about authors’ expectations based on previous research.
The research is sound and provides good information about the state of art. It is worrying from my point of view that out of 19 studies, only one of the selected studies is from Europe and it was on 1994. Authors may contribute more comments in the discussion section about the lack of sound research in this valuable field.
Author Response
Thank you for your serious and constructive comments on our manuscript. According to your suggestion, the manuscript has been revised as a letter to editor. The revisions we have made are as follows:
Point 1: The introduction section may benefit from more information about risky sexual behaviors in both adolescence and adult life. For instance, information about possible causes and theories about may help readers to understand the relevance of the study and could help authors to elaborate the discussion section.
Response 1: Thank you for your constructive and helpful suggestion. In the revised paper, the relevant information has been mentioned in page 2, line 47 to 49.
Point 2: With this regard, the objectives subsection in the introduction should include hypothesis about authors’ expectations based on previous research.
Response 2: Thank you for your constructive and helpful suggestion. In the revised paper, the hypothesis has been mentioned in page 2, line 68 to 70.
Point 3: The research is sound and provides good information about the state of art. It is worrying from my point of view that out of 19 studies, only one of the selected studies is from Europe and it was on 1994. Authors may contribute more comments in the discussion section about the lack of sound research in this valuable field.
Response 2: Thank you for your constructive and helpful suggestion. In the revised paper, the relevant discussion has been mentioned in page 14, line 323 to 325.
